# Conservation and Expansion of Transcriptional Factor Repertoire in the *Fusarium oxysporum* Species Complex

**DOI:** 10.3390/jof9030359

**Published:** 2023-03-15

**Authors:** Houlin Yu, He Yang, Sajeet Haridas, Richard D. Hayes, Hunter Lynch, Sawyer Andersen, Madison Newman, Gengtan Li, Domingo Martínez-Soto, Shira Milo-Cochavi, Dilay Hazal Ayhan, Yong Zhang, Igor V. Grigoriev, Li-Jun Ma

**Affiliations:** 1Department of Biochemistry and Molecular Biology, University of Massachusetts Amherst, Amherst, MA 01003, USA; 2Department of Energy Joint Genome Institute, Lawrence Berkeley National Laboratory, University of California Berkeley, Berkeley, CA 94720, USA; 3Department of Plant and Microbial Biology, University of California Berkeley, Berkeley, CA 94598, USA

**Keywords:** *Fusarium oxysporum* species complex, transcription factors, TFome, accessory chromosome, conservation, expansion

## Abstract

The *Fusarium oxysporum* species complex (FOSC) includes both plant and human pathogens that cause devastating plant vascular wilt diseases and threaten public health. Each *F. oxysporum* genome comprises core chromosomes (CCs) for housekeeping functions and accessory chromosomes (ACs) that contribute to host-specific adaptation. This study inspects global transcription factor profiles (TFomes) and their potential roles in coordinating CC and AC functions to accomplish host-specific interactions. Remarkably, we found a clear positive correlation between the sizes of TFomes and the proteomes of an organism. With the acquisition of ACs, the FOSC TFomes were larger than the other fungal genomes included in this study. Among a total of 48 classified TF families, 14 families involved in transcription/translation regulations and cell cycle controls were highly conserved. Among the 30 FOSC expanded families, Zn2-C6 and Znf_C2H2 were most significantly expanded to 671 and 167 genes per family including well-characterized homologs of Ftf1 (Zn2-C6) and PacC (Znf_C2H2) that are involved in host-specific interactions. Manual curation of characterized TFs increased the TFome repertoires by 3% including a disordered protein Ren1. RNA-Seq revealed a steady pattern of expression for conserved TF families and specific activation for AC TFs. Functional characterization of these TFs could enhance our understanding of transcriptional regulation involved in FOSC cross-kingdom interactions, disentangle species-specific adaptation, and identify targets to combat diverse diseases caused by this group of fungal pathogens.

## 1. Introduction

The fungal species complex of *Fusarium oxysporum* (FOSC) has been used as a model to study cross-kingdom fungal pathogenesis. Members within FOSC can cause devastating fusarium wilt diseases among economically important crops [1,2,3,4,5,6,7,8,9,10,11,12] and is listed among the top five most important plant pathogens [8]. With strong host specificity, plant pathogenic *F. oxysporum* strains are further grouped as *formae speciales* [13]. For instance, tomato pathogens are named *F. oxysporum* f. sp. *lycopersici*; cotton pathogens are *F. oxysporum* f. sp. *vasinfectum* [11], and banana pathogens are *F. oxysporum* f. sp. *cubense* [10]. Recently, members within FOSC have also been reported to be responsible for fusariosis, the top emerging opportunistic mycosis [1,7,12], and fusarium keratitis, one of the major causes of cornea infections in the developing world and the leading cause of blindness among fungal keratitis patients [14,15].

Comparative genomics studies on this cross-kingdom pathogen revealed that the FOSC genomes, both human and plant pathogens, are compartmentalized into two components: the core chromosomes (CCs) and accessory chromosomes (ACs). While CCs are conserved and vertically inherited to execute essential housekeeping functions, horizontally transmitted ACs are lineage- or strain-specific and are related to fungal adaptation and pathogenicity [1,7,12].

To coexist and function within the same genome, ACs and CCs coordinate their gene expression. One intriguing cross-regulation example, reported in the reference genome of *F. oxysporum* f. sp. *lycopersici Fol*4287, includes transcription factors Sge1 (SIX Gene Expression 1), Ftfs, and virulent factors SIX (Secreted in Xylem) proteins. Sge1 is a highly conserved CC-encoding TF. By name definition, Sge1 regulates the expression of SIX proteins [16,17]. The *Fol*4287 genome encodes an AC-encoding Ftf1 protein and one CC-encoding Ftf2 (Ftf1 CC homolog) [17]. Constitutive expression of either *Ftf1* or *Ftf2* induced the expression of effector genes [17]. Furthermore, it has been documented that DNA binding sites of Sge1 and Ftf1 are enriched among the cis-regulatory elements of in planta transcriptionally upregulated genes [17]. Another example of CC and AC cross-talking is the alkaline pH-responsive transcription factor PacC/Rim1p reported in *F. oxysporum* clinical strains [18]. In addition to the full-length *PacC* ortholog (*PacC_O*), located on a CC, the clinical isolate NRRL32931 genome encodes three truncated *PacC* homologs, named *PacC_a, PacC_b*, and *PacC_c* in ACs [18].

To thoroughly understand the coordination of the crosstalk between genome compartments and their contribution to the cross-kingdom fungal pathogenesis, this study compared the repertoire of TFs (i.e., TFome) among 15 *F. oxysporum* and 15 other ascomycete fungal genomes. Remarkably, we discovered a strong positive correlation (*y* = 0.07264*x* − 190.9, *r*^2^ = 0.9361) between the number of genes (*x*) and TFome size (*y*) of an organism. Primarily due to the acquisition of ACs, we observed increased TFome sizes among the FOSC genomes. All TFs were organized into 48 families based on the InterPro classification of proteins. Fourteen families involved in transcription/translation regulations and cell cycle controls were highly conserved. Thirty families, accounting for ¾ of all families, were expanded to various degrees among the FOSC genomes. Unique TF expansions driven by ACs include members of the Zn2-C6 fungal-type (Zn2-C6) and Zinc Finger C2H2 (Znf_C2H2) families. This comparative study highlighted conserved regulatory mechanisms. The signature of conservation established the foundation to study the various impacts of additional AC TFs on existing regulatory pathways. In combination with the existing expression data, this study provides insights into the fine-tuning of environmental adaptation performed by this group of diverse organisms in order to engage in cross-kingdom interactions with different hosts.

## 2. Materials and Methods

### 2.1. Generation of Fungal TFomes

The annotation pipeline is briefly summarized in Appendix A. The fungal proteomes of 30 strains were downloaded from the JGI MycoCosm portal [19]. Protein annotation was performed using InterProScan/5.38–76.0 (https://www.ebi.ac.uk/interpro/search/sequence/, accessed on 7 February 2023) [20]. Annotations of proteins that putatively serve as TFs were filtered out using a table containing InterPro terms related to transcriptional regulatory functions summarized in the literature [21,22], with further addition by manual curation (Appendix A). Orthologous analysis was conducted with OrthoFinder 2.5.4 (https://github.com/davidemms/OrthoFinder, accessed on 7 February 2023) [23] to probe the orthologs of functionally validated TFs (Appendix A) in *Fusarium*.

### 2.2. RNA-Seq Analysis

The RNA-Seq datasets were previously described [24,25] and deposited by those authors to the NCBI Short Read Archive with accession number GSE87352 and to the ArrayExpress database at EMBL-EBI (www.ebi.ac.uk/arrayexpress, accessed on 7 February 2023) under accession number E-MTAB-10597, respectively. For data reprocessing, reads were mapped to reference genomes of Arabidopsis [annotation version Araport11 [26]], Fo5176 [27], Fo47 [28], and Fol4287 [29] using HISAT2 version 2.0.5 [30]. Mapped reads were used to quantify the transcriptome by StringTie version 1.3.4 [31], at which step TPM (transcript per million) normalization was applied. Normalized read counts were first averaged per condition, transformed by log2 (normalized read count + 1) and Z-scaled. This was then visualized in pheatmap (version 1.0.12).

### 2.3. Genome Partition

The genome partition results for chromosome-level assemblies were retrieved from previous reports for Fol4287 [29], FoII5 [32], Fo5176 [27], and Fo47 [28]. Fo47 has a clear genome partition with 11 core chromosomes and one accessory chromosome, therefore serving as the reference for the genome partition of other *F. oxysporum* genomes. MUMmer/3.22 was applied to align scaffolds of genome assemblies against 11 core chromosomes of the reference genome Fo47 using default parameters. The scaffolds aligned to the core chromosomes of Fo47 with a coverage larger than 5% were annotated as core scaffolds. The rest of the scaffolds were partitioned as accessory scaffolds. Genes residing on the core and accessory scaffolds were annotated as the core and accessory genes, respectively.

### 2.4. Phylogenetics Analysis

Protein sequences were aligned via MAFFT/7.313 [33]. The iqtree/1.6.3 [34,35] was run on the sequence alignment to generate the phylogeny (by maximum likelihood method and bootstrapped using 1000 replicates) [36]. Visualization was conducted via the Interactive Tree of Life [37] to produce the phylogram. OrthoFinder 2.5.4 [23] was used for orthogroup determination. To build a species phylogram, 500 randomly selected conserved proteins (single-copy orthologs) were aligned first. The alignment was then concatenated, and the phylogeny was determined and visualized using the above methods.

### 2.5. Expansion Index Calculation

To understand genome regulation among FOSC, we developed two expansion index scores. The first uses two yeast lineages as the baseline (*EI_y_*):$${EI}_{y}=\frac{Average\;number\;of\;TFs\;in\;FOSC+1}{Average\;number\;of\;TFs\;in\;yeasts+1}$$

In the second index score (*EI_f_*), we directly compared *F. oxysporum* with its *Fusarium* relatives to calculate the expansion index as follows:$${EI}_{f}=\frac{Average\;number\;of\;TFs\;in\;FOSC+1}{Average\;number\;of\;TFs\;in\;FOSC\;sister\;species+1}$$

## 3. Results

### 3.1. FOSC TFome Expansion Resulted from the Acquisition of ACs

We compared 30 ascomycete fungal genomes (Figure 1 and Table 1) including 15 strains within the FOSC, nine sister species close to *F. oxysporum*, two yeast genomes (*Saccharomyces cerevisiae* and *Schizosaccharomyces pombe*), and four other filamentous fungal species (*Neurospora crassa*, *Aspergillus nidulans*, *Aspergillus acristatulus*, and *Magnaporthe oryzae*). To maintain consistency, the protein sequences for all of these genomes were retrieved from the MycoCosm portal [19].

For a comprehensive TFome annotation, we used InterProScan (IPR) terms associated with fungal transcriptional regulation [21,22] and curated a mapping with updated IPR classification (interproscan version: 5.38–76.0) [52]. In addition, we searched the IPR classification of the protein families and obtained all other terms related to the transcriptional regulation activity. This resulted in 234 TF-related IPR terms (Appendix A). Since most terms were initially defined in the mammalian systems, fungal genomes included in this study were associated with 71 out of the 234 TF-related IPR terms (Appendix A, Materials and Methods, and Appendix A for the annotation pipeline). After removing 13 terms for redundancy (two terms describing the identical domain) and 10 terms for minimal presentation (<4 among the 30 genomes), this comparative TFome study focused on 48 IPR terms in 27,967 TFs (Appendix A). Notably, a quarter of these terms were not reported to be affiliated with fungal transcriptional regulation by either Park et al. (2008) [21] or Shelest (2017) [22] (Appendix A).

Comparing the total number of protein-coding genes (*x*) and the total number of TFs (*y*) within the same genome, we observed a strong positive correlation (*y =* 0.07264*x −* 190.9, *r*^2^ = 0.9361) (Figure 2A). FOSC TFomes were larger than other genomes included in this study, with an average of 1144 TFs per genome (Figure 2A, Table 1). After partitioning each FOSC genome into core and accessory regions (see Section 2.3 for details), we observed a positive correlation between the number of TFs encoded in the accessory chromosomal region of each strain (defined as accessory TFs hereafter) with the size of the accessory genomes (Mb) (*y* = 17.239*x* + 3.553) (Appendix A). This suggests that accessory chromosomes contribute directly to the expanded TFome.

To understand the genome regulation among the FOSC, we developed an expansion index score using two yeast lineages as the baseline (*EI_y_*):$${EI}_{y}=\frac{Average\;number\;of\;TFs\;in\;FOSC+1}{Average\;number\;of\;TFs\;in\;yeasts+1}$$

Based on this index value, we classified TF families into three major groups (Table 2 and Appendix A). Group 1 contained 14 TF families with an expansion score of 1, indicating high conservation. Group 2 included four families with an index score below 1, reflecting some level of gene family contraction. Group 3 contained 30 families with an expansion index greater than 1, indicating gene expansion.

### 3.2. Conserved TF Families That Are Primarily Associated with General/Global Transcription Factors

Fourteen TF families accounted for 30% of our annotated TF families. Most of these fourteen families had a single ortholog in all genomes included in this study (Figure 2B; Table 2 and Appendix A), suggesting their functional conservation across the *Ascomycota*. These 30% conserved TF families accounted for less than 2% of the total TFomes. Annotation based on *S. cerevisiae* and other model organisms suggested their involvement in transcription/translation regulation and cell cycle control.

#### 3.2.1. Transcription/Translation Regulation

Nine TF families were annotated to be related to transcription and translational regulation including TATA box-binding protein (TBP), TBP-associated factors (TAFs), RNA polymerase II elongation regulator Vps25, and CCAAT-binding factors (CBFs) related to ribosomal biogenesis.

One of the most conserved TF families, transcription initiation TBP binds directly to the TATA box to define the transcription start and initiate transcription facilitated by all three RNA polymerases. In fact, the function of TBP is so conserved that the yeast homolog can complement *TBP* mutations in humans [53,54]. Seven conserved TF families are classified as transcription positive/negative regulators, and transcription elongation factors. TAF12 and TAF_II_28 are parts of the transcription factor TF_II_D complex. Interacting with TBP, TAFs form the TF_II_D complex and positively participate in the assembly of the transcription preinitiation complex [55]. Similarly, TF_II_H works synergistically with TF_II_D to promote the transcription [56]. In contrast, negative cofactor 2 (Ncb2) inhibits the preinitiation complex assembly [57]. Other factors include the CNOT1, a global regulator involved in transcription initiation and RNA degradation [58], and Vps72/YL1, which contributes to transcriptional regulation through chromatin remodeling, as reported in yeast [52,59]. Vps25 is a subunit of the ESCRT-II complex, which binds to the RNA polymerase II elongation factor to exert transcriptional control in mammalian systems [60]. One TF family is suggested to be involved in translational regulation. CCAAT box is a common cis-acting element found in the promoter and enhancer regions of genes in the eukaryotes [61,62]. CBFs are necessary for 60S ribosomal subunit biogenesis and are therefore involved in translational control [63,64,65]. This family including Noc3, Noc4, and Mak21 in *S. cerevisiae* had three members in each genome, and a clear single-copy orthologous relationship could be observed for each member (Appendix A).

#### 3.2.2. Cell Cycle Control

Five TF families were related to cell cycle control including cell cycle progression, DNA repair, and machinery/cell integrity maintenance.

One conserved TF family, APSES-type HTH, was reported to be involved in cell cycle control and crucial to development [66]. Every genome included in this study encoded four copies of the APSES-type HTH gene (Appendix A) that formed four clades of single-copy orthologs in all genomes except yeasts. Genes in Clade 1 included StuA homologs. As a target of the cyclic AMP (cAMP)-dependent protein kinase A (PKA) signal transduction pathway, StuA was reported to be involved in dimorphic switch [67,68], fungal spore development, and the production of secondary metabolites [69]. Genes in Clade 2 and Clade 3 included *S. cerevisiae* Swi4 and Swi6, which were reported to form a protein complex regulating cell cycle progression from G1 to the S phase [70] as well as meiosis [71]. Genes in Clade 4 included homologs of *S. pombe* Bqt4, anchoring telomeres to the nuclear envelope [72].

The conserved TF family, DTT, represented by the *S. cerevisiae* homolog Itc1, is recognized as a subunit of the ATP-dependent Isw2p-Itc1p chromatin remodeling complex and is required for the repression of early meiotic gene expression during mitotic growth [73].

The other conserved TF family, RFX, was reported to be involved in DNA repair. Each strain encoded two orthologous copies, except for *F. venenatum* encoding two copies within the RFX1 clade (Appendix A). Being a major transcriptional repressor of DNA-damage-regulated genes in *S. cerevisiae*, Rfx1 functions in DNA damage repair and replication checkpoint pathways [74]. In *F. graminearum*, Rfx1 was reported to be essential in maintaining the genome integrity [75]. The other copy, Rsc9 in *S. cerevisiae*, was reported to be a member of the chromatin structure-remodeling complex RSC, which is involved in transcription regulation and nucleosome positioning [76,77].

The conserved TF family NFYA was reported to bind to the CCAAT box. All strains maintained a single copy of this family. Its yeast homolog, Hap2, has been reported to induce the expression of mitochondrial electron transport genes [78] and its *F. verticillioides* homolog NFYA Hap2 was reported to be essential for fungal growth and the virulence on maize stalks [79].

The MADS MEF2-like TF family including *S. cerevisiae* Rlm1 was reported to be a component of the protein kinase C-mediated MAP kinase pathway involved in maintaining cell integrity [80]. Having a paralog from the whole genome duplication in *S. cerevisiae*, Rlm1 was detected as a single copy gene in all filamentous fungi included in this study. Its member in *F. verticilioides,* Mef2 has been reported to play a vital role in sexual development [81].

### 3.3. Gene Family Contractions in FOSC Partially Caused by Whole Genome Duplication in Yeast

We detected an expansion score of less than 1 for four TF families, MATalpha_HMGbox, NOT4, MADS_SRF-like, and HSF (heat shock factor), reflecting some level of gene family contraction among members of FOSC compared to the two yeast genomes (Appendix A).

TF family MATalpha_HMGbox including *S. cerevisiae* mating type protein alpha 1 has been reported to be a transcription activator that activates mating-type alpha-specific genes [82]. Reflecting the potential heterothallic mating strategy, all *F. oxysporum* Mat1-1 type strains contained this TF, but not the Mat1-2 strains, even though sexual reproduction has not been observed in FOSC [83].

TF family NOT4 was reported to be a component of the multifunctional CCR4-NOT complex, a global transcriptional repressor of the RNA polymerase II transcription [84]. Most genomes included in this study encoded one copy of this TF family, but some filamentous fungal genomes including *A. nidulans*, *F. redolens*, *F. oxysporum* strains NRRL26365, MRL8666, and PHW726 lost it. The functional implication of this loss remains to be discovered.

The contractions of the other two TF families, MADS SRF-like and HSF, were primarily caused by the whole genome duplication in yeast. In contrast to the contraction at the global scale, both TF families expanded among some FOSC strains when compared to other filamentous fungi (Appendix A).

As reported in *M. oryzae,* MADS SRF-like TF, essential for the transcriptional regulation of growth-factor-inducible genes [85], is important for microconidium production and virulence in host plants [86]. Due to the event of whole genome duplication, the *S. cerevisiae* genome contained two copies of this TF family, while all filamentous genomes encoded a single copy. However, we detected an average of 2.73 copies among the phytopathogenic FOSC strains. There were six copies in the genome of Fo5176, a pathogen of Brassicaceae plants including *A. thaliana* (Appendix A).

TF family HSF has been reported to activate the production of heat shock proteins that prevent or mitigate protein misfolding under abiotic/biotic stresses [87]. The *S. cerevisiae* genome contained five copies of the HSF TF family, while all non-FOSC filamentous fungi had three copies. Members of FOSC exhibited some level of expansion to four or five copies (Fo47:4, Fol4287: 5, II5: 4, HDV274: 4, and Fo5176: 4), with 1–2 copies encoded in ACs. Phylogenetically, all HSF TFs were clustered into three major clades, named as Skn7, Sfl1, and Hsf1 (Figure 3A,B). All AC-encoding HSFs were phylogenetically close to Hsf1 (Figure 3A). Based on the study in *M. oryzae*, the family Sfl1 is essential for vegetative growth, conidiation, sexual reproduction, and pathogenesis [88]. Based on a study in *F. graminearum*, the family Skn7 is involved in regulating the oxidative stress response and is essential for pathogenicity [89]. Our expression data generated during the plant colonization [24] supported the involvement of all three core genes during plant colonization (Figure 3C). However, the Hsf1 accessory copies of these two strains were distinct, as the Fo47 AC-encoding Hsf1 was upregulated and the Fo5176 AC-encoding Hsf1 was downregulated, post inoculation (Figure 3C), suggesting that their distinct regulatory function is involved in these two distinct interactions.

### 3.4. Significant FOSC TFome Expansion Driven by a Few Exceedingly Expanded TF Families

#### 3.4.1. Gain-of-Function among Filamentous Ascomycete Fungi

Three TF families, CP2 (*EI_y_* = 2.73), HTH_AraC (*EI_y_* = 2), and HTH_Psq (*EI_y_* = 3.53), were absent in both yeast genomes, suggesting a gain of function among filamentous ascomycete fungi (Appendix A). TF family CP2 was studied in animal and fungal kingdoms with a function related to differentiation and development [90]. Both HTH_AraC and HTH_Psq are part of the helix-turn-helix (HTH) superfamily. HTH_AraC was first reported in bacteria as a positive regulator regulating the arabinose operon [91,92,93]. HTH_Psq, as part of the eukaryotic Pipsqueak protein family, has been reported in vertebrates, insects, nematodes, and fungi to regulate processes involved in cell death [94]. Most FOSC genomes encoded a single copy of HTH_AraC, while the number of HTH_Psq-containing proteins ranged from 0 to 9 in the FOSC and 0 to 3 in other *Fusarium* genomes. Since the HTH_Psq domain also exists in transposases [94], and ACs in FOSC are transposon-rich, it remains to be studied whether proteins containing the Psq domain are bona fide TFs.

#### 3.4.2. Seven Exceedingly Expanded TF Families

Among the families containing minimally one yeast ortholog, seven TF families had expansion indices greater than 2 (Table 2 and Figure 2B) including Zn2-C6 (*EI_y_* = 15.09), bZIP (*EI_y_* = 5.80), Znf_C2H2 (*EI_y_* = 4.15), Homeobox (*EI_y_* = 2.28), PAI2 (*EI_y_* = 3.42), NDT80 (*EI_y_* = 3.47), and bHLH (*EI_y_* = 3.48). Based on the number of increments, the most significantly expanded TF families were Zn2-C6 (44 in yeasts versus 671 in FOSC) and Znf_C2H2 (40 in yeasts versus 167 in FOSC) (Figure 2C and Appendix A). Because of their large expansion, these seven families accounted for more than 75% of the total TFome. All seven families exhibited a gradual expansion, following the pattern yeasts < non-*Fusarium* filamentous fungi < non-FOSC *Fusarium* < FOSC (Appendix A). Annotating large TF families could be challenging. Here, we described some examples based on the literature.

Zn2-C6, a fungal TF family [95], was detected as the most significant expanded TF family, reaching over 600 members among the FOSC genomes and accounting for more than half of the total TFome. Able to form a homodimer, this group of TFs are also able to bind to the specific palindromic DNA sequence through direct contact with the major groove of the double-stranded DNA molecules [95]. The versatility of this group of TFs can be achieved by domain shuffling and by changing the nucleotide binding specificity. In addition to the well-documented Ftf1 [17,96,97,98,99], five additional TFs within this family were reported in *F. oxysporum* including Ctf1 [100], Ctf2 [100], Fow2 [101], XlnR [102] and Ebr1 [103]. Their functions were reported to be involved in the development, metabolism, stress response, and pathogenicity.

Znf_C2H2 was reported to be the most common DNA-binding motif found in the eukaryotic transcription factors [104]. Five reported *F. oxysprum* TFs are Czf1 [105], Con7-1 [106], PacC [18,107], ZafA [108], and St12 [109,110]. Classified in the Znf_C2H2 family, PacC has been linked to the fungal virulence in both plant and human hosts [18,107].

The other five families were bZIP, Homeobox, PAI2, Ndt80, and bHLH. The bZIP domain contains a region for sequence-specific DNA binding followed by a leucine zipper region required for dimerization [111]. Three *F. oxysporum* bZIP TFs were reported including Atf1 [112], Hapx [113], and MeaB [114], all of which are important for fungal pathogenicity. Homeobox is a DNA binding motif with a helix-turn-helix structure. In *S. pombe,* a homeobox-domain containing protein Phx1 was reported to be a transcriptional coactivator involved in yeast fission. In *M. oryzae,* the homeobox-domain containing protein Hox played roles in conidiation and appressorium development [115]. The TF family PAI2 is involved in the negative regulation of protease synthesis and sporulation of the *Bacillus subtilis* [116]. The TF family Ndt80 is essential for completing meiosis in *S. cerevisiae* [117,118] and *Ustilago maydis* [119] by promoting the expression of sporulation genes for the fulfillment of meiotic chromosome segregation [120]. The TF family bHLH forms a superfamily of transcriptional regulators found in almost all eukaryotes and is involved in diverse developmental processes [121]. In *F. graminearum*, a bHLH-domain containing protein Gra2 was reported to regulate the biosynthesis of phytotoxin gramillin [122], while a bHLH-domain containing protein SreA in *Penicillium digitatum* is required for anti-fungal resistance and full virulence in citrus fruits [123].

#### 3.4.3. Other Families

The other 20 TF families (expanded but with *EI_y_* ≤ 2) accounted for 20% of the TFome, with an average 9.6 copies in each genome examined (Appendix A).

Four TF families were functionally linked to chromatin remodeling including Bromodomain (*EI_y_* = 1.52), CBFA_NFYB (*EI_y_* = 1.35), Znf_RING-CH (*EI_y_* = 1.11), and ARID (*EI_y_* = 1.25). The TF family Bromodomain contained Spt7. As a crucial part of the SAGA complex in yeast, Spt7 has been reported to recognize the acetylated lysines of histones and eventually lead to chromatin unwinding [124]. The CBFA_NFYB domain was found in proteins (e.g., *S. cerevisiae* Dls1) that regulate RNA polymerase II transcription by controlling the chromatin accessibility (e.g., telomeric silencing) [125]. The TF family Znf_RING-CH also had a functional connection to chromatin modification (e.g., *S. cerevisiae* Rkr1) [126]. The domain ARID, a 100 amino acid motif, has been found in many eukaryotic TFs [127] such as Swi1 in *S. cerevisiae*, playing an important role in chromatin remodeling. This domain is also required to transcribe a diverse set of genes including some retrotransposons [128,129].

TFs belonging to the Ste12 family are only found in the fungal kingdom. Except for *S. pombe*, every genome encodes one copy. Binding to a DNA motif that mediates pheromone response, Ste12 TFs were reported to regulate fungal development and pathogenicity [130] and are involved in mating and pseudohyphae formation [131]. In *F. oxysporum,* Ste12, downstream of the Fmk1-mediated MAPK cascade, is involved in the control of invasive growth and fungal virulence [110].

Among the others, the Znf_NFX1 domain has been found in NK-X1, a repressor of the human disease-associated gene HLA-DRA [132]. The HMG_box (high mobility group box) in *S. cerevisiae* is seen in three proteins: Spp41, which is involved in the negative expression regulation of spliceosome components [133]; Nhp6a, which is required for the fidelity of some tRNA genes [134]; and Ixr1, a transcriptional repressor that regulates hypoxic genes [135]. Fep1, an example of Znf_GATA, was reported to be a transcriptional repressor involved in the regulation of some iron transporter genes under high iron concentrations [136]. *S. cerevisiae* Mbf1, belonging to Cro/C1-type HTH, is a transcriptional coactivator [137].

### 3.5. Orthologous Survey of TF Families That Were Manually Curated

To further understand expanded TFs and their impacts on transcriptional regulation, we curated a list of 102 TFs reported in the literature focusing on *F. oxysporum*, *F. graminearum,* and other phytopathogenic fungi (Appendix A and examples as described in the previous section). Compared to this list of curated TFs using Orthofinder, we defined 80 orthologous groups among the *Fusarium* genomes (Appendix A). Sixty-two out of the 80 orthogroups were identified using the above IPR-annotated pipeline including 17 in Zn2C6, nine in Znf_C2H2, and one containing both the Znf_C2H2 and Zn2-C6 domains (Appendix A). This helps add to the functional annotation of these large TF families while also adding additional annotation to 18 TF families (Appendix A), accounting for 32 genes per genome (3% of average *Fusarium* TFome size). These newly annotated TFs include homologs of those without a domain annotation (e.g., disordered proteins *F. oxysporum* Ren1 [138], *M. oryzae* Som1 [139], and homologs of those with noncanonical TF domains such as Ankyrin_rpt and WD40_repeat).

We then directly compared *F. oxysporum* with its *Fusarium* relatives to calculate the expansion index as follows:$$EI_{f}=\frac{Average\;number\;of\;TFs\;in\;FOSC+1}{Average\;number\;of\;TFs\;in\;FOSC\;sister\;species+1}$$

The *EI_f_* ranged from the highest score of 3.54 (Fug, AreA_GATA) to the lowest score of 0.5 (Fox1, Fork_head) (Appendix A). Among these 80 orthogroups, 36 groups were conserved (*EI_f_* = 1), with one gene per genome. Ten of these conserved groups were functionally validated in *F. oxysporum* (Appendix A). Twenty four groups had scores less than 1, while 20 groups had a score greater than 1 (Table 3 and Appendix A). Expanded groups included Fug1 (AreA_GATA, *EI_f_* = 3.54), Cos1 (Znf_C2H2, *EI_f_* = 2.8), Ftf1/Ftf2 (Zn2-C6, *EI_f_* = 2.7), Ebr1/Ebr2 (Zn2-C6, *EI_f_* = 2.5), and Ren1 (disordered, *EI_f_* = 2). We also identified PacC (*EI_f_* = 1.57) as the second most expanded group within the highly expanded Znf_C2H2 family. We will further discuss these six groups (Table 3).

Both Ftf1/Ftf2 and Ebr1/Ebr2, belonging to the Zn2-C6 family, contributes directly to fungal virulence [3,17,97]. The deletion of AC-encoding Ftf1 reduced the pathogenicity of *F. oxysporum* f. sp. *phaseoli* [97], highlighting the direct functional involvement of AC TF in virulence. In *Fol*, deleting either Ftf1 (AC encoding) or Ftf2 (CC encoding) reduced the fungal virulence [96,152]. Constitutive expression of either Ftf1 or Ftf2 induced the expression of effector genes [17]. The core copy Ftf2 was conserved among all *Fusarium* species, and the AC copy Ftf1 was only found in *F. oxysporum* and *F. redolens* (Figure 4). Ebr1 had multiple homologs in *F. oxysporum*, but a single copy in *F. graminearum* [103]. The *F. oxysporum* genome had three AC-encoding paralogs: Ebr2, Ebr3, and Ebr4. Interestingly, these AC-encoding paralogs are regulated by core copy Ebr1 [103]. It is worth noting that the Ebr2 coding sequence driven by the Ebr1 promoter was able to rescue the Ebr1 knockout mutation, indicating some functional redundancy of this family.

Both Cos1 and PacC are part of the Znf_C2H2 family. In *M. oryzae*, Cos1 was reported to be involved in conidiophore development [112] and functions as a negative regulator reducing fungal pathogenicity [153]. PacC has been reported as an important pH-responsive TF in *F. oxysporum* [18,107]. This TF family was expanded in clinical strains, showing an average accessory copy number of 3.7 of FOSC, whereas the non-clinical strains showed an average accessory copy number of 0.5. All of the *Fusarium* relatives’ genomes examined only contained a single copy of the core PacC. Our previous study using one *F. oxysporum* clinical isolate revealed that the expression of all PacC genes can be induced with a pH shift from 5.0 to 7.4 (the mammalian physiological pH), indicating a potential role in host adaptation [18]. Interestingly, the induction of AC-encoding PacC genes was CC-encoding PacC gene-dependent as the induction disappeared in the CC-encoding PacC knockout mutant, further supporting a cross-talk between the core and accessory TFs. Similar to EBR1, the expression of the AC-encoding PacC genes was much lower than that of the CC-encoding PacC gene, and knockouts of one AC PacC gene affected a small subset of genes compared with the CC PacC knockout, which had a broader effect on the cellular processes [49].

Fug1 has a role in pathogenicity (maize kernel colonization) and fumonisin biosynthesis in *F. verticillioides* [147]. The deletion of Fug1 increased the sensitivity to the antimicrobial compound 2-benzoxazolinone and to hydrogen peroxide, suggesting its role in mitigating stresses associated with the host defense [147]. Neither CC nor AC-encoding copies of these two genes were experimentally examined in FOSC. Ren1, a disordered protein without IPR functional domain, was expanded with a *EI_f_* score of 2 among the FOSC. However, the only reported study on its function is in *F. oxysporum* f. sp. *melonis*, regulating the development of conidiation [138].

### 3.6. Transcriptome Analysis to Probe the Essential TFs during Host Colonization

To understand the functional importance of FOSC TFs, we took advantage of two recently reported transcriptomics datasets [24,25] including pathogenic interactions (Fo5176 infecting Arabidopsis and Fol4287 infecting tomato) and endophytic interactions (Fo47 colonizing Arabidopsis) (Appendix A).

By examining patterns of expression (Appendix A), we found that almost all genes within the conserved category (58 out of 60) (Group 1) were consistently expressed (TPM > 1 across all conditions), supporting their general roles in controlling life processes. Within the expanded category (Group 3), the proportion of genes being consistently expressed ranged from 41% to 59% for core TFs and only from 5% to 16% for AC-encoding TFs. With a less strict filter (TPM > 1 at minimum 1 condition), we found that all genes within the conserved category were expressed. Within the expanded category, 93% of core TFs and between 49% and 67% of AC-encoding TFs were expressed. The significant increase in the AC-encoding TFs with a lower stringency further supported their conditional involvement in niche adaptation.

We further reviewed the expression patterns of the reported TFs in Fol4287 (Appendix A). Out of the 27 TFs encoded on the core genome, 18 were upregulated (defined by upregulation under at least three out of four in planta conditions compared to the axenic growth) during plant colonization, which is consistent with their reported roles in pathogenicity. The AC-encoding Ftf1 has been reported to play essential functions in fungal pathogenicity [96]. Of the ten accessory Ffs, eight were upregulated during plant colonization.

Using a higher stringency filter, selecting TFs upregulated under all in planta conditions, we searched for: (1) conserved core TFs that may be related to plant colonization and (2) expanded AC TFs that could be related to host-specific pathogenicity. In the Fol4287, Fo5176, and Fo47 genomes, 95, 62, and 44 core TFs were upregulated during plant colonization, respectively. Among them, ten copies were highly conserved (Appendix A) as they were single-copy orthologs across all 15 *F. oxysporum* strains including Fow2 and Sfl1. Fow2, a Zn2C6 TF, is required for full virulence but not hyphal growth and conidiation in *F. oxysporum* f. sp. *melonis* [101]. Sfl1 was reported to be essential for vegetative growth, conidiation, sexual reproduction, and pathogenesis in *M. oryzae* [88].

Fol4287, Fo5176, and Fo47 contained 29, 34, and nine upregulated accessory TFs, respectively, including *Ftf1* and *Ren1* (Figure 4 and Appendix A). Ftfs have been reported to play an essential role in the pathogenicity in Fol4287, although their involvement in other interactions has not. The Fol4287 genome encoded 10 accessory *Ftf*s and eight were upregulated during plant colonization. The Fo5176 genome encoded six accessory *Ftf*s, but only one copy was upregulated during plant colonization. Interestingly, eight upregulated Fol4287 and one upregulated Fo5176 *Ftf*s were clustered together (Figure 4). The unique expansion with regulatory adaptation (i.e., fine-tuned expression regulation) seemed to be restricted to Fol4287 and not the other pathogenic strain, Fo5176. In contrast, the Fo5176 genome encoded seven accessory *Ren1* TF and two were upregulated during plant colonization (Figure 4), while the Fol4287 genome had only one accessory *Ren1* not involved in host colonization. While functional validation is needed, strain-specific expansion followed by fine-tuned expression regulation when infecting host species exists and likely contributes to host-specific pathogenicity.

## 4. Discussion

For a soilborne pathogen with strong host specificity like FOSC, the adjustment of growth and cell cycle control in response to environmental cues is likely to be essential for survival. Expanded TF families likely contribute to the enhanced functions related to niche adaptation as these TF families play important roles in transmitting external and internal signals and regulating complex cellular signaling responses to the sensed stimuli. Therefore, it is not surprising that the genomes of FOSC had larger TFome sizes than the other fungi included in the study. The expansion of TFs among the FOSC resulted in a positive correlation between the total number of proteins and the size of the fungal TFome, which was also observed in other instances [22].

A total of 14 TF families that control the global transcriptional event such as TBP are highly conserved within the ascomycete fungal lineages. Conserved regulatory mechanisms revealed through this study suggest that the plant colonization process could be a common process among FOSC strains regardless of their host-specific pathogenesis. This notion is also supported by recent studies that highlighted the ability of FOSC strains as root colonizers regardless whether of they cause disease or function as endophytes [25,154].

In contrast to these stable TFs, 30 families were expanded to various degrees, and the most significant expansions occurred in the Zn2-C6 and Znf_C2H2 TF families among the FOSC genomes. The number of Zn2-C6 TFs increased significantly (with the highest expansion score) and made up most of the TFs (56.7%) found within the FOSC TFome. For example, Ftf1, a TF belonging to Zn2-C6 and is involved in tomato pathogenicity, was most significantly expanded to 10 copies of accessory Ftfs in the tomato pathogen Fol4287 genome. Eight out of 10 were induced during plant colonization.

The unique expansion of some TFs, driven by ACs, may provide clues as to host-specific interactions. Acquiring additional TFs will modify existing regulatory pathways, and this will require fine-tuning existing networks for this group of organisms to successfully adapt to different hosts under diverse environments. A previous survey of kinome (the complete set of protein kinases encoded in an organism’s genome) among the FOSC and other Ascomycetes also revealed a positive correlation between the size of the kinome and the size of the genome [48], identical to what we reported here for TFomes. As kinases and TFs are key regulators that modulate all important signaling pathways and are essential for the proper functions of almost all molecular and cellular processes, strong correlations between kinome and TFome suggest the ordered recruitment and establishment of ACs among FOSC genomes.

This realization further emphasizes the importance of additional functional studies. Reverse genetics is a powerful tool in defining the functional importance of a TF. For example, TF Ren1, a disordered protein, was identified by genetic and molecular characterization [138]. This TF is most significantly expanded (seven copies of accessory *Ren*s) in the Arabidopsis pathogen Fo5176 genome and is involved in plant colonization. High throughput approaches such as chromatin immunoprecipitation sequencing (CHIP-Seq) and DNA affinity purification sequencing (DAP-Seq) [155] can be used to profile the cis-regulatory elements globally for a better understanding of transcriptional regulation in the fungal model *F. oxysporum*. Gene regulatory networks [156,157] can add more resolution to these complex regulatory processes. However, the ultimate understanding of the regulatory roles of each TF will come from careful molecular and biochemical characterization.

Our study offers a comprehensive look at the regulation from the evolutionary perspective while also providing an easily implemented computational pipeline to compare TFs and other functional groups in fungi. A better understanding of functions of TFs will not only inform *Fusarium* biology [158], but can also be extrapolated to other filamentous fungal systems.

## Figures and Tables

**Figure 1 jof-09-00359-f001:**
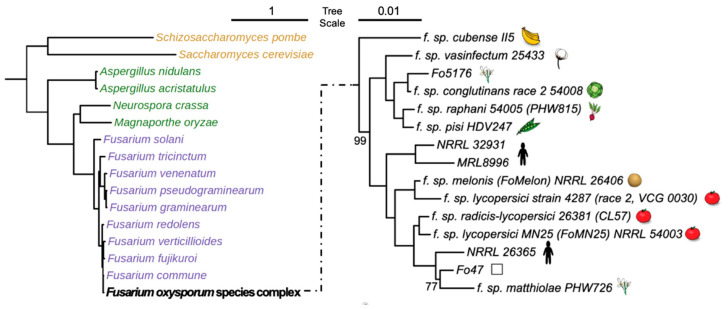
Phylogeny of fungal genomes included in this study. Both left and right phylograms were constructed by the concatenated alignment of randomly selected 500 single-copy orthologous proteins, followed by the maximum likelihood method with 1000 bootstraps. Left shows a phylogram of FOSC (represented by the reference genome Fol4287) together with the other 15 ascomycetes. The right shows a phylogram of members within FOSC, rooted by *F. verticillioides* (not shown). Only bootstrap values not equal to 100 are shown.

**Figure 2 jof-09-00359-f002:**
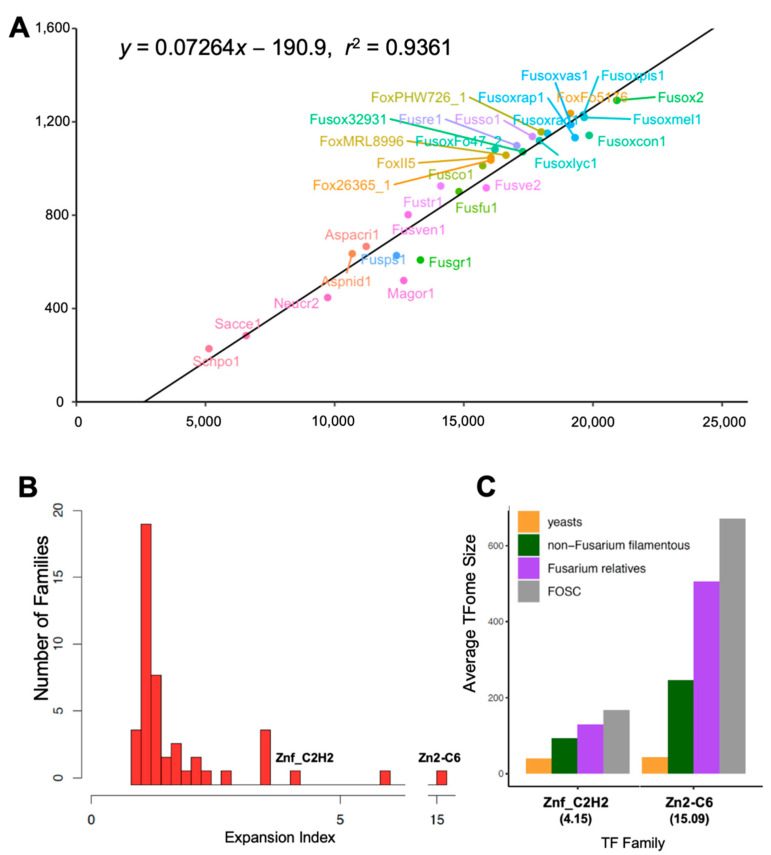
TFome conservation and variation among ascomycete fungi: baseline description. (**A**) There is a positive correlation between the number of protein-coding genes and TFome size of an organism. JGI fungal genome identifiers were used as labels. (**B**) Histogram illustrates the distribution of expansion indices among different families. (**C**) Average number of TFs of the two most drastically expanded families (Znf_C2H2 and Zn2-C6) within each genome set. Genome Set 1 (G1) includes two yeast genomes (*S. cerevisiae* and *S. pombe*). Genome Set 2 (G2) includes four filamentous fungal species (*N. crassa*, *A. nidulans, A. acristatulus,* and *M. oryzae*). Genome Set 3 (G3) includes nine sister species close to *F. oxysporum*. Genome Set 4 (G4) includes 15 FOSC genomes.

**Figure 3 jof-09-00359-f003:**
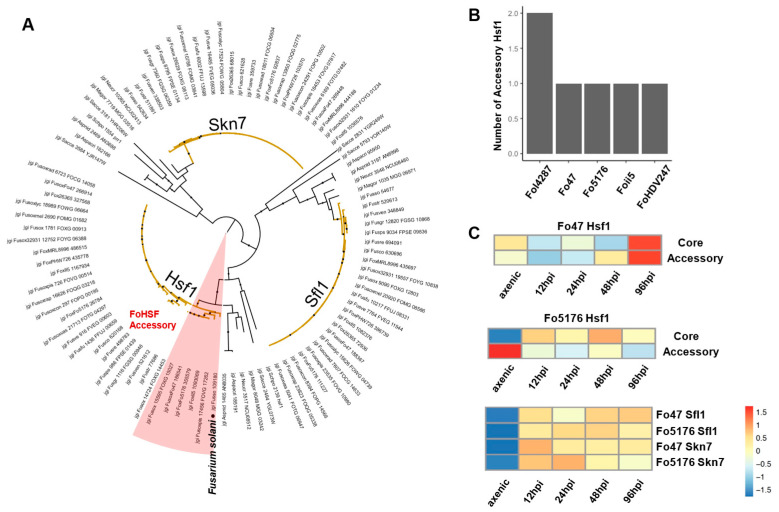
Evolutional trajectory of heat shock factors (HSFs) suggests genome expansion and adaptation. (**A**) Phylograms of HSFs were constructed by the maximum likelihood method with 1000 bootstraps. Branches of *Fusarium* HSFs are colored in yellow. Accessory HSFs of FOSC are shaded in red. (**B**) Number of accessory HSFs in some FOSC genomes. (**C**) Expression of *HSF* genes during plant colonization (hpi indicates hours post inoculation) compared to axenic growth. Transcriptome data were previously described in Guo et al. 2021 [24]. See Section 2 for details of the data reprocessing and visualization.

**Figure 4 jof-09-00359-f004:**
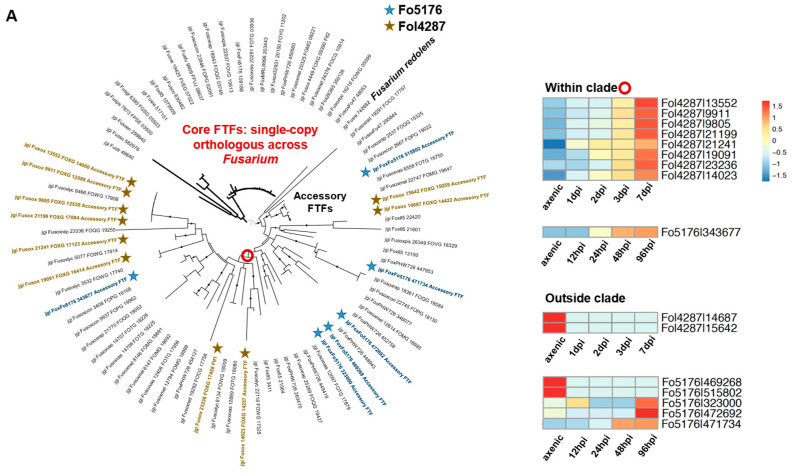
Unique expansion of some TFs, driven by ACs, may provide clues to host-specific adaptation. RNA-Seq data were previously described [24,25]. (**A**) Ftf1, the TF involved in tomato pathogenicity was most significantly expanded (10 copies of accessory FTFs) in the tomato pathogen Fol4287 genome and the expression of eight out of 10 were induced during plant colonization. (**B**) Ren1 was the most significantly expanded (seven copies of accessory RENs) in the Arabidopsis pathogen Fo5176 genome, and two of them were induced during plant colonization.

**Table 1 jof-09-00359-t001:** Fungal genomes used in this study.

Fungal Species or Strains	MycoCosm Identifier	Genome Size (MB)	No. of Genes	TFome Size	Host	Reference
*Saccharomyces cerevisiae*	Sacce1	12.07	6575	284		[38]
*Schizosaccharomyces pombe*	Schpo1	12.61	5134	228		[39]
*Aspergillus nidulans*	Aspnid1	30.48	10,680	635		[40]
*Aspergillus acristatulus*	Aspacri1	32.59	11,221	666		[41]
*Neurospora crassa*	Neucr2	41.04	9730	447		[42]
*Magnaporthe oryzae*	Magor1	40.49	12,673	520	Rice	[43]
*Fusarium solani*	Fusso1	52.93	17,656	1137	broad hosts	[44]
*F. pseudograminearum*	Fusps1	36.33	12,395	627	Wheat	[45]
*F. graminearum*	Fusgr1	36.45	13,321	608	Wheat	[46]
*F. venenatum*	Fusven1	37.45	12,845	802		[44]
*F. tricinctum*	Fustr1	43.69	14,106	925	Broad hosts	[44]
*F. verticillioides*	Fusve2	41.78	15,869	917	Corn	[29]
*F. fujikuroi*	Fusfu1	43.83	14,813	901	Broad hosts	[47]
*F. redolens*	Fusre1	52.56	17,051	1098	Broad hosts	[44]
*F. commune*	Fusco1	48.37	15,731	1012	Broad hosts	[44]
*F. oxysporum* f. sp. *cubense* (II5)	FoxII5	49.43	16,048	1047	Banana	[32]
*F. oxysporum* f. sp. *radicis*-*lycopersici* (CL57)	Fusoxrad1	49.36	18,238	1151	Tomato	[48]
*F. oxysporum* Fo47 (Fo47)	FusoxFo47_2	50.36	16,207	1082		[28]
*F. oxysporum* f. sp. *lycopersici* (MN25)	Fusoxlyc1	48.64	17,931	1119	Tomato	[48]
*F. oxysporum* NRRL26365 (NRRL26365)	Fox26365_1	48.46	16,047	1036	Human	[49]
*F. oxysporum* f. sp. *melonis* (FoMelon)	Fusoxmel1	54.03	19,661	1219	Melon	[2]
*F. oxysporum* f. sp. *lycopersici* (Fol4287)	Fusox2	61.36	20,925	1292	Tomato	[29]
*F. oxysporum* NRRL32931 (NRRL32931)	Fusox32931	47.91	17,280	1072	Human	[18]
*F. oxysporum* MRL8996 (MRL8996)	FoxMRL8996	50.07	16,631	1057	Human	[18]
*F. oxysporum* f. sp. *matthiolae* (PHW726)	FoxPHW726_1	57.22	17,996	1157	Brassica	[50]
*F. oxysporum* f. sp. *vasinfectum* (FoCotton)	Fusoxvas1	52.91	19,143	1189	Cotton	[48]
*F. oxysporum* f. sp. *pisi* (HDV247)	Fusoxpis1	55.19	19,623	1229	Pea	[51]
*F. oxysporum* f. sp. *raphani* (PHW815)	Fusoxrap1	53.5	19,306	1132	Brassica	[48]
*F. oxysporum* f. sp. *conglutinans* (PHW808)	Fusoxcon1	53.58	19,854	1142	Brassica	[48]
*F. oxysporum* Fo5176 (Fo5176)	FoxFo5176	67.98	19,130	1236	Arabidopsis	[27]

**Table 2 jof-09-00359-t002:** Expansion index (*EI_y_*) of 48 TF families.

IPR	Term	*EI_y_*
Group 1		
IPR000814	TBP	1
IPR003228	TFIID_TAF12	1
IPR004595	TFIIH_C1-like	1
IPR006809	TAFII28	1
IPR042225	Ncb2	1
IPR008570	Vps25	1
IPR008895	Vps72/YL1	1
IPR007196	CNOT1	1
IPR005612	CBF	1
IPR001289	NFYA	1
IPR018004	APSES-type HTH	1
IPR003150	RFX	1
IPR033896	MADS_MEF2-like	1
IPR018501	DDT	1
Group 2		
IPR006856	MATalpha_HMGbox	0.8
IPR039515	NOT4	0.9
IPR033897	MADS_SRF-like	0.95
IPR000232	HSF	0.98
Group 3		
IPR003163	Tscrpt_reg_HTH_APSES-type	1.04
IPR001766	Fork_head	1.05
IPR011016	Znf_RING-CH	1.11
IPR001965	Znf_PHD	1.11
IPR009071	HMG_box	1.12
IPR004181	Znf_MIZ	1.24
IPR001606	ARID	1.25
IPR000679	Znf_GATA	1.3
IPR001005	SANT/Myb	1.32
IPR000818	TEA/ATTS	1.33
IPR003120	Ste12	1.33
IPR003958	CBFA_NFYB	1.35
IPR001083	Cu_fist	1.37
IPR000967	Znf_NFX1	1.4
IPR006565	Bromodomain	1.52
IPR001387	Cro/C1-type_HTH	1.6
IPR001841	Znf_RING	1.64
IPR000571	Znf_CCCH	1.74
IPR001878	Znf_CCHC	1.83
IPR010666	Znf_GRF	2
IPR018060	HTH_AraC *	2
IPR001356	Homeobox	2.28
IPR007604	CP2 *	2.73
IPR007396	PAI2	3.42
IPR024061	NDT80	3.47
IPR011598	bHLH	3.48
IPR007889	HTH_Psq *	3.53
IPR013087	Znf_C2H2	4.15
IPR004827	bZIP	5.8
IPR001138	Zn2-C6	15.09

* Indicates the families without a presence in yeasts.

**Table 3 jof-09-00359-t003:** Ortholog copy number and expansion index (*EI_f_*) of the characterized and expanded TFs in *F. oxysporum*.

TF	Reported Species	References	Family	Overlap *	Average_Fo	Average_ Non-Fo	*EI_f_*
Ftf1/Ftf2	*F. oxysporum*	[96]	Zn2-C6	Yes	4.80	1.11	2.75
Ebr1/Ebr2	*F. oxysporum*	[103]	Zn2-C6	Yes	5.27	1.56	2.45
Znf1	*M. oryzae*	[140]	Zn2-C6	Yes	6.47	2.78	1.98
Ctf2	*F. oxysporum*	[141]	Zn2-C6	Yes	2.93	1.33	1.69
Fow2	*F. oxysporum*	[101]	Zn2-C6	Yes	2.07	1.00	1.53
Dep6	*A. brassicicola*	[142]	Zn2-C6	Yes	0.93	0.67	1.16
Pf2	*A. brassicicola*	[143]	Zn2-C6	Yes	1.20	1.00	1.10
Art1	*F. verticilioides*	[144]	Zn2-C6	Yes	1.00	0.89	1.06
Clta1	*C. lindemuthianum*	[145]	Zn2-C6	Yes	1.07	1.00	1.03
Fhs1	*F. graminearum*	[146]	Zn2-C6	Yes	1.07	1.00	1.03
Cos1	*M. oryzae*	[112]	Znf_C2H2	Yes	1.80	0.00	2.80
PacC	*F. oxysporum*	[107]	Znf_C2H2	Yes	2.13	1.00	1.57
Fug1	*F. verticillioides*	[147]	AreA_GATA	No	7.27	1.33	3.54
Ren1	*F. oxysporum*	[138]	disordered	No	3.00	1.00	2.00
Tri10	*F. graminearum*	[148]	Fun_TF	No	1.13	0.33	1.60
Ltf1	*B. cinerea*	[149]	Znf_GATA	Yes	4.00	2.44	1.45
Ndt80	*U. maydis*	[119]	NDT80	Yes	1.73	1.11	1.29
Hap3p	*F. verticillioides*	[147]	CBFA_NFYB	Yes	1.33	1.00	1.17
Sod1	*F. oxysporum*	[150]	SOD_Cu_Zn	No	1.47	1.22	1.11
Prf1	*F. oxysporum*	[151]	HMG_box	Yes	1.07	1.00	1.03

* Column ‘Overlap’ indicates whether orthologous mapping probed families were already included in our domain-based TF annotation.

## Data Availability

Unprocessed RNA-Seq data were retrieved from the public source as described in Section 2.2. All processed data supporting our results and conclusions are available in the Appendix A.

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
