# Peer review of "Conservation and Expansion of Transcriptional Factor Repertoire in the Fusarium oxysporum Species Complex"

_jof, 2023, doi:10.3390/jof9030359_

Round 1

Reviewer 1 Report

The authors present a very extensive work on the transcriptional Factors in the Fusarium oxysporum species complex. This work tackles a great variety of species and transcriptional factor families. However, there are certain aspects should be revised.

The English of the manuscript should be reviewed and improved.

Throughout the manuscript there are several abbreviations that need to be written in their full form when they appear for the first time.

The authors state that they developed an expansion index score. This information and respective formulas should be included in materials and methods.

When the authors talk about the results on “Conserved TF families that are primarily associated with general/global transcription factors” very often the data seems to be well illustrated by figure S3. I think it might useful to have this image in the main text and not as a supplemental figure. Moreover, the image and respective legend should be more detailed.

In the results section, for instance in “3.5. Orthologous survey of TF families that were manually curated”, there are several statements that are more suitable for the discussion section. I advise the authors to revise the results and discussion sections and to rearrange them to separate the statement of the results from the discussion of these results and respective integration with the literature.

Specific suggestions:

Line 55: The authors state that “A few characterized transcription factors (TFs) coordinate the crosstalk between CCs and ACs, two compartments.” Please clarify the context of “two compartments” in this sentence, as it is not clear. Do the authors mean that the TFs coordinate the crosstalk between CCs and ACs in two compartments?

Lines 85-87: The authors state that “In combination with existing expression data, this study may provide clues to the fine-tuning of networks in the environmental adaptation of this group of diverse organisms to engage in complex cross-kingdom interactions with different hosts.” However, I advise the authors to be more assertive about their conclusions from this study: either the study provides clues or it doesn’t.  

Figure 2: The images in this figure, especially part A, should be amplified because it is hard to read its content. Moreover, the labelling in B and C should be more thorough. In B all the different families should be labelled and in C is lacking a legend for the genome sets and the y axis.

The authors state that they developed an expansion index score. This information and respective formula should be included in materials and methods.

Line 347: That authors indicate that “Based on both high expansion index and large number increment, we considered Zn2-C6 and Znf_C2H2 as the most significantly expanded families”. How can this affirmation be supported by statistics? Please clarify this and include all the relevant information in Materials and Methods section.

Line 387: Since P. digitatum is appearing for the first time in the text it should be written in the full form. Also, the whole sentence regarding this species seems to be out of context because it is also the first time in the manuscript that SreA is mentioned. Please give a context to this sentence. Lastly, this discussion should be present in the discussion section and not in the result section.

Table 3 should be presented in a more harmonious way. For instance, there are column headers that have interrupted words. This makes it harder to read the table.

Figure 4: The threes should be amplified to make the figure easier to read.

Reviewer 2 Report

The authors Yu et al present interesting work on the transcription factors in filamentous fungi, with specific focus on the FOSC. The work combines both new analyses and a review of the literature on many different TF families. This manuscript will likely serve as an important reference for researchers studying transcription factors in filamentous fungi.  A few minor suggestions are noted below.

165 - Does the number of genes on the x-axis include transposons?

208 - suggest changing ‘as’ to ‘that’

260 - suggest changing ‘conservativeness’ to ‘conservation’

268 - I didn’t really understand what was meant by this header until reading the subsequent section. I would suggest removing ‘Minimal’, as it’s not immediately clear it it’s referring to ‘gene families’ or ‘contractions’.

282 - Did you try using BLAST with the gene sequence from another F. oxysporum genome against the three that apparently did not have it?

312 - Should this be post inoculation? Infection would occur at some point after inoculation, but probably not immediately.

314 - I think you mean section 3.6. It strikes me that this use of transcription to probe function is not unique to TFomes and is quite common in the literature. I think the novelty of the results is apparent without this sentence.

339 - Not sure what is meant by ‘among others’

354 - Is there a relationship between Zn2C6 transcription factors and virulence in any other fungus? Do you have hypotheses as to why this particular family is so exceptionally expanded and related to virulence in FOSC?

389 - capitalize ‘o’?

422 - It’s not totally clear why TFs were identified in two different ways and the lists analyzed separately rather than being combined and analyzed together. 

499 - was should be were

542 - what is meant by ‘restricted to the same strain’?

568 - This sentence appears fragmented 

Figures in general - much of the text on axes is very small, please consider enlargening

Figure 1 - the legend implies that every single branch has a bootstrap value of 100 except for two on the FOSC partition. Is this true? This would be unusual compared to bootstrap values in other FOSC trees built from single copy orthologues.

Figure 2 - text on figure 2 is very small. Also suggest adding the name of the color corresponding to each genome set.

Figure 4 - what is meant by ‘within clade’ and ‘outside clade’? I don’t see mention of these terms in the legend or main text. I would also suggest reminding readers that the plant hosts are different for Fo5167 and Fol4287 in the datasets used. 

Table 3 would benefit from footnotes to explain what is meant by ‘overlap’, potentially other headers
